# Fostering culture of care for early career researchers—Building a trustful environment: Insights from a German perspective

Fernando Gonzalez-Uarquin[1]*, Fabienne Ferrara[2], Nadine Baumgart[1], Jan Baumgart[1], Sabine Juliane Bischoff[3]

1 TARCforce3R Center, University Medical Center of the Johannes Gutenberg University, Mainz, Germany, 2 Consulting and Training in Animal Research (ConscienceTrain), Berlin, Germany, 3 Institute of Laboratory Animal Science, Medical Faculty, Heinrich-Heine-University, Duesseldorf, Germany

☯ These authors contributed equally to this work.
* dfernandogonzalez@gmail.com, fernando.gonzalez@uni-mainz.de

## Abstract

Early-career researchers (ECRs) play a key role in conducting animal experiments in academic research. However, they face considerable challenges, including poor working conditions, and inadequate strategies for managing distress. These difficulties are often amplified in animal research, where a lack of consensus on the 3Rs (replacement, reduction, and refinement), challenges to navigate complex regulations and ethical dilemmas can further complicate the situation. These challenges not only jeopardize the well-being of both animals and researchers, but also undermine the quality of scientific work, potentially driving ECRs out of academia. This paper explores the relationship between 3Rs training, communication, and stress-coping mechanisms used by ECRs in Germany and provides recommendations to address the challenges. To that end, we employed a dual-method approach: surveying ECRs to gather their perspectives on working with laboratory animals, communication challenges, and stress management, while also conducting a workshop to exchange insights with experienced professionals who engage with ECRs in their daily roles. Our results highlight the many difficulties faced by ECRs working on animal experiments, which range from organizational challenges to the practical implementation of the experiments. These insights emphasize the need for a holistic strategy that includes direct engagement with ECRs and the development of policies focused on their overall well-being. Our recommendations advocate for improved communication strategies, recognition of cultural differences, implementation of peer coaching and mentoring programs, and strengthening institutional support systems particularly with regard to the conduct of animal experiments. To conclude, this research calls for a transformative shift towards a more supportive and inclusive environment for ECRs, harmonizing scientific progress with ethical standards.

**Data availability statement:** All relevant data are within the paper and its Supporting information files.

**Funding:** The author(s) received no specific funding for this work.

**Competing interests:** The authors have declared that no competing interests exist.

## Introduction

Early Career Researchers (ECRs) actively contribute to advancements while bringing fresh perspectives that fuel scientific breakthroughs in academia. Still, ECRs encounter instrumental and emotional challenges. The ECRs' voices are increasingly vocal on pressing issues such as deteriorating working conditions, chronic stress, restrictive temporary contract legislation (such as the "*WissZeitVG*" in Germany), and the challenges of ensuring the reproducibility of scientific findings [1–5]. The ECRs' discomfort is reflected in the results of a recent survey conducted by the German Centre for Higher Education Research and Science Studies (DZHW), where more than half of all scientists in Germany express serious consideration of leaving academia [5]. The DZHW survey could be exacerbated as it did not consider, particularly, the field of animal research, where ECRs not only grapple with the ethical and scientific complexities of their work but also contend with the emotional strain of caring for, researching with, and inevitably euthanizing animals [6,7].

A critical barrier to addressing these challenges is the lack of effective internal communication between ECRs and their co-workers and supervisors. The lack of effective internal communication jeopardizes ECRs' mental health and animal well-being, undermining their creativity, innovation, and leadership potential [8]. This threatens the future of academia by either decreasing the quality of science or leading to job quitting. Recognizing this problem, our endeavors must identify communication bottlenecks with ECRs and strategies for fostering better work relationships and engagement. Improved internal communication also means better communication regarding animal studies licenses, planning and conducting experiments, and a better understanding and application of the 3Rs (replacement, reduction, and refinement [9]) tools. Ultimately, this enhances their overall well-being in animal care contexts [10,11]. The challenge lies in cultivating an environment where scientific pursuit aligns seamlessly with the advancements of scientific 3Rs, ethical responsibility, and the well-being of researchers and animals.

In this study, we identified the status of internal communication between ECRs and their colleagues and supervisors across Germany, focusing on their understanding and application of the 3Rs, as well as their stress perception and coping in animal experimentation. To this end, we applied two methodologies: 1) an in-depth survey targeting ECRs, inquiring into nuanced facets of their professional challenges, including perspectives on 3R training, communication dynamics, and coping mechanisms for inherent stressors, and 2) a workshop involving experienced professionals intimately familiar with the challenges faced by ECRs, who provided valuable insights derived from practical experiences and keen field observations. We outline strategies obtained from both activities for implementing communication and stress-coping strategies to enhance the quality of life for both ECRs and animals.

## Materials and methods

To gain deeper insights into the current work environment and the needs of ECRs in Germany, we launched an online survey as a first step in studying this population

in animal research. To promote a supportive future work culture, we also conducted a workshop on the culture of care, communication, and sharing for better well-being and science at the GV-SOLAS (German Society for Laboratory Animal Science) and IGTP (Interest Group Animal Caretakers) Conference 2023. The Commission on Ethics and Scientific Integrity of the Karl Landsteiner University of Health Sciences, Austria, evaluated the proposal and found that according to their institutional standards and the provisions of the Helsinki Declaration in its most recent version there are no formal requirements or ethical concerns that would require approval by an ethical committee or to preclude further consideration for publication in a scientific journal.

## Survey description

We implemented an online survey (S1 Table) with 26 questions using the LimeSurvey platform (www.limesurvey.de), licensed by the University Medical Center of Johannes Gutenberg University Mainz. Access to the survey was set for three months (August – October 2023). The collected data is archived on access-protected servers of the University Medical Center of Johannes Gutenberg University Mainz under the guidelines of good scientific practice (GSP). The survey was designed to be completely anonymous and voluntary. All procedures adhered to EU data protection regulations. All data from this survey will be stored for ten years under GSP and then deleted, provided there are no legal obligations to retain the data. The IP addresses of the participants are anonymized; that is, no IP addresses can be recorded or stored. Participants were informed and agreed that information collected by this service could be used for publication. No approval was sought from the animal protection authority, as there were no interactions between the researchers and animals during this study.

Means of diffusion across Germany were our institutional mailing lists and the member mailing list of the GV-SOLAS association. The target was ECRs in their late PhD or early postdoc positions. The survey required about 10 minutes to complete. We divided the questions into five groups:

*Group 1- Demographic data:* We asked participants four questions regarding general demographics.

*Group 2- Stress Coping and Resilience:* In eight questions, we collected information about the sources of stress for ECRs, stress management and tools to cope with stress when working with animal experiments.

*Group 3- Communication: We asked six questions to gauge the quality of communication and cooperation of ECRs with their colleagues and supervisors and to devise* strategies for addressing communication challenges regarding animal experiments.

*Group 4- Training:* We asked seven questions about the training that ECRs received in 3Rs, animal-free methodologies and stress management. We also collected information about resources, tools or training programs ECRs use for training.

*Group 5- Final comments:* We asked for additional comments and observations in this final question.

With the exception of the demographic questions, a 5-point Likert scale was employed for all other items. Some of the questions were multiple some of them single choice questions. Further details can be found in S1 Table.

We explored quantitative data (closed-choice responses) using descriptive statistics. Qualitative data (Open comments and feedback) underwent thematic analysis to identify recurring themes, extract meaningful insights, and provide recommendations. Data visualization was made using Microsoft Excel (Microsoft Corporation, 2019). S2 Table provides the results of all the questions.

## Workshop description

The workshop was conducted as part of the *60th Annual Meeting of the Society for Laboratory Animal Science (GV-SOLAS) and the 21st Advanced Training Course of the IGTP*, held in Mainz, Germany, between September 6–8, 2023. The workshop had 12 voluntary (registered) participants split into four groups (three people each). The small-group design was intended to maximize the sharing of experiences and ideas to foster a supportive work culture. Attendees

were mostly senior staff (older 40 years old) who had constant communication with ECRs. Participants shared their individual experiences, clarifying that their views did not represent the positions of their respective organizations.

The workshop was entitled *"Culture of Care: Communicating and Sharing with Early Career Researchers for Better Well-Being and Science"* and lasted 90 minutes. The workshop was divided into five parts. Each contained a short keynote presentation, followed by group work or group activity time focused on experience and best practice ideas, with a final concluding discussion. Questions to be answered and topics to be discussed in the workshop were presented in a PowerPoint presentation (Stress-coping mechanisms, communication and training). The results of the discussions were presented directly, noted on post-its, and explained by a representative of the group. At the end of the workshop, an online tool was used to capture the key take-home messages.

*Part 1—Work-life challenges in science as Early Career Researchers:* In this introductory section, we presented common challenges that ECRs must cope with while conducting animal experiments, along with the potential consequences for the well-being of animals, humans and science (e.g., poor animal handling, compassion fatigue syndrome, burnout and reduced scientific quality).

*Part 2- Early Career Researchers and Animal Care:* We focused on the importance of knowledge and skills for ECRs in performing animal experiments and recognizing and understanding the concept of the 3Rs.

*Part 3- Communication with Early Career Researchers:* We shared the preliminary results of the survey published in this paper. We highlighted trust and cultural and intercultural aspects driving communication (e.g., the power of asking questions).

*Part 4- Stress Coping and Resilience:* We defined mental stress, coping and resilience and discussed work-related stress. In the following step, we depend on the principal aspects driving stress in animal experiments while presenting preliminary results from the survey published in this paper.

*Part 5- Conclusion and Wrap-up:* We summarized the four groups and agreed on a shared responsibility to improve the situation of the ECRs involved in animal experiments. We concluded the workshop by obtaining the participants' takeaways.

## Results

### Demographic characterization of the survey

The demographic profile of our study participants showcased a diverse and inclusive sample. A total of 124 ECRs took part in the survey. Regarding age distribution, most respondents fell within the 31–40 years range (60.48%), followed by those in the 18–30 years range (39.52%). Gender representation reflected a slight predominance of females (65.32%) over males (33.06%), with a small percentage opting not to disclose their gender (1.61%). The academic background of participants was predominantly rooted in biomedical sciences (62.10%), highlighting a substantial proportion of individuals with expertise in human health-related fields. Biology emerged as the second most prevalent field of study, encompassing 20.97% of the respondents, while veterinary sciences and other disciplines (most related to biomedical science) constituted 8.87% and 8.06%, respectively. In terms of professional experience, our participants exhibited varied durations of working with animals, with 6.45% reporting less than one year, 17.74% one to two years, 28.23% three to five years, and 47.58% more than five years. This comprehensive demographic overview enhances the richness of our dataset, ensuring the inclusion of diverse perspectives and experiences within the study cohort.

**ECR: Internal communication regarding animal experiments.** The communication proficiency among ECRs and their supervisors is depicted in Fig 1. In the question 13, we asked participants to rate their communication with their supervisor regarding animal experiments. The response options were *"Poor," "Below average," "Average," "Good,"* and *"Excellent."* Our results indicated that 24.2% of the surveyed ECRs reported suboptimal communication *("poor"+"below average"*) categorized as *"Poor"* in Fig 1A. 23.39% of the participants reported average communication with their supervisors, and 50% of ECRs reported good or excellent communication. We also asked ECRs about their perceived

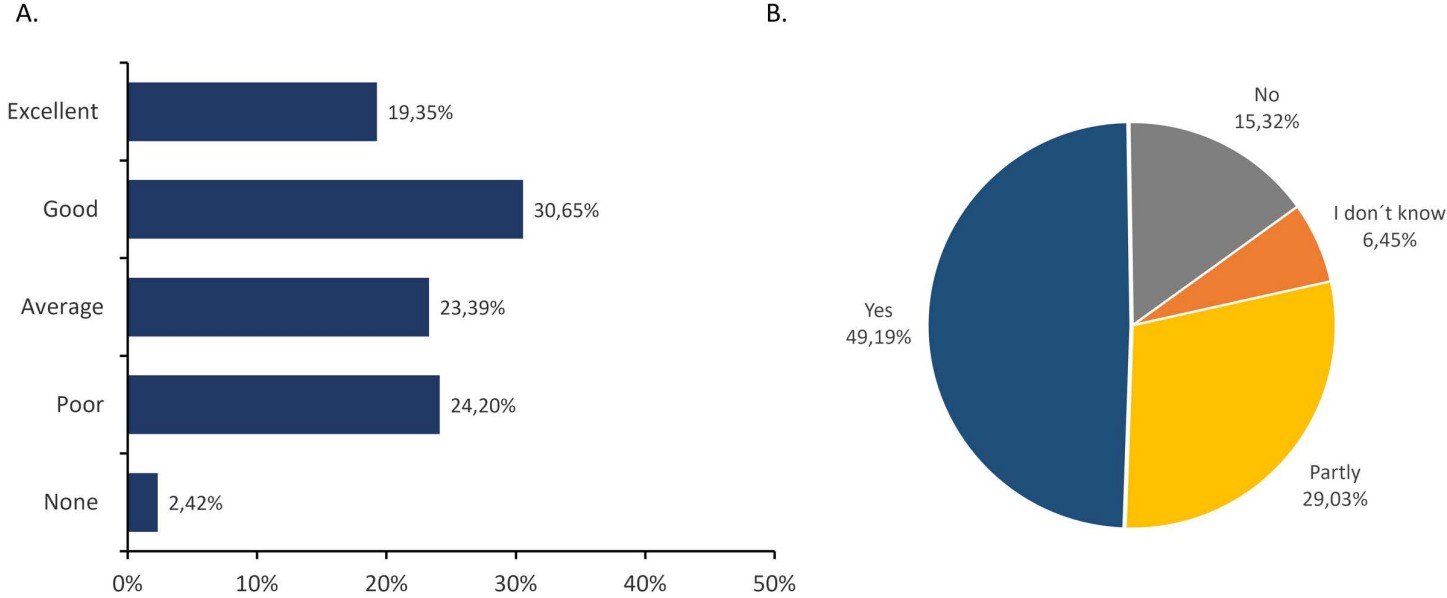

**Fig 1. Level of communication between the ECRs and their supervisors regarding animal experiments.** In these questions (number 13 and 15 of the survey), participants were asked: A: *"How would you rate the communication with your supervisor regarding animal experiments?"* B: *"Do you feel valued by your supervisor?"* Both questions reflected a similar pattern reflecting about half of the surveyed ECRS perceived optimal communication with their supervisors.

value provided by their supervisors (Fig 1B): 49,19% reported feeling valued, 15,32% reported not feeling valued, while 29,03% and 6,45% said *"partially"* and *"I do not know"*, respectively.

The findings concerning communication dynamics between ECRs and their research groups or co-workers are illustrated in Fig 2. In this question, we asked participants to rate their communication with their co-workers regarding animal experiments. The response options were *"Poor," "Below average," "Average," "Good,"* and *"Excellent."* 18.54% of those surveyed reported suboptimal communication *("poor"* + *"below average")* categorized as *"Poor."* 22.58% responded that their communication was average, whereas 58.87% of the participants reported good or excellent communication.

We ranked the responses to question 18: *"What strategies would you envision for coping with the challenges and limitations in communicating with your co-workers and supervisor about animal experiments?"* According to the given answers, we categorized the aspects that ECRs considered essential for improvement to facilitate communication with co-workers and supervisors regarding animal experiments (Table 1).

### ECR: Training and education

The results of training and education in animal care and the 3Rs are shown in Fig 3. We asked about regularity of training in animal care and 3Rs. Answer options were provided with scores (1 for "never" and score 5 for "regularly") Although some participants (4.03%) state that they receive no training in these areas, a substantial proportion receive repeated or regular training. Asking whether these training courses are voluntary or compulsory and whether only theoretical or practical knowledge and skills are taught provided more detailed insights (we mention them in the Discussion section).

**ECR: Stress coping and resilience.** To evaluate the status of key stress factors in ECRs' careers, we asked the participants about their sources of stress. Answer options were given: 1.) Experimenting with animals (e.g., handling and sacrificing animals); 2.) The uncertainty of your professional future; 3.) Pressure to obtain results and publish your work. Answer options were provided with scores (1 for "never" and score 5 for "very often"). The results are shown in Fig 4. For

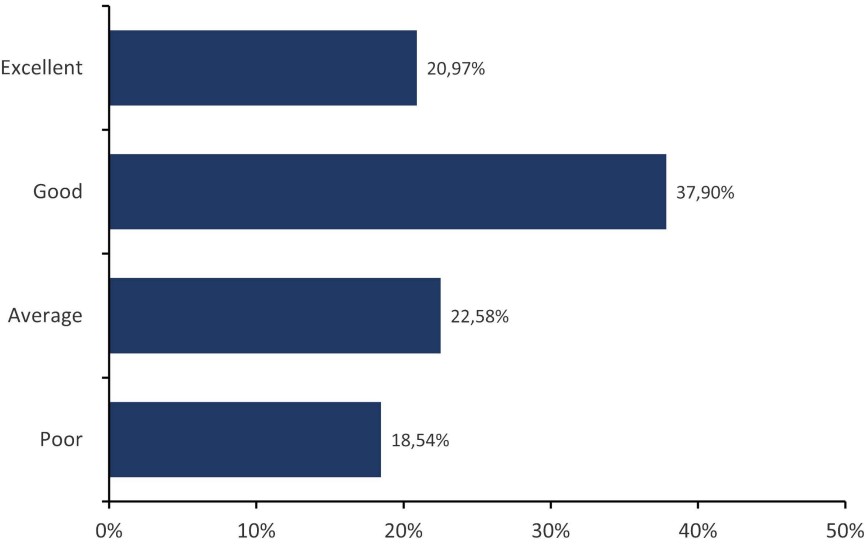

**Fig 2. Level of communication between the ECRs and their research groups (co-workers) regarding animal experiments.** In this question (number 17 of the survey), participants were asked: *"Please rate the level of teamwork and collaboration within your research group in experiments that involve animal experiments."* Nearly one-fifth of respondents indicated suboptimal communication experiences. Approximately one-fifth of ECRs reported experiencing average communication, while a significant majority of ECRs, consisting of 37.90% and 20.97%, reported good and excellent communication, respectively.

the first answer option, 37.50% of participants chose score 2 (rarely); 41.41% chose for the second answer option score 5 (very often), and 29.03% chose score 4 (often) for the third answer option.

To gain insights into institutional support or personal strategies on stress coping mechanisms and resilience strengthening, we asked the participants in question 8 about the tools provided by their institution to manage stress (Fig 5) and in question 9 about their mechanisms to cope with stressful situations (Fig 6). Answer options for question 8 were "yes," "no," and "I don't know." Answer options for question 9 were: *"Seeking support from colleagues and superiors," "Practicing self-care activities such as exercise, meditation, and using apps for mindfulness," "Setting boundaries between work and personal life," "I don't have specific coping mechanisms,"* and *"Others."* Results of question 8 on knowledge about institutional support in providing tools to manage stress and improve resilience. 40.32% of participants chose the answer option *"no,"* 45.97% chose the answer *"I don't know,"* and only 13.71% of participants *chose "yes."* Results of question 9 on personal cope with stress and challenges related to daily work. 59.68% of participants chose *"setting boundaries between work and personal life,"* followed by *"seeking support from colleagues and superiors"* (55.65% of participants) and *"practicing self-care activities such as exercise, meditation, and using apps for mindfulness"* (50.81% of participants). Participants also chose the option *"I don't have any specific coping mechanisms"* (16.13% of participants) and the option *"others"* (5.65%). In both questions, participants were allowed to provide a written note (highlighted in the survey as *"make a comment on your choice here"*) about the actual institutional support offers or personal strategies. These comments are listed in Tables 2 and 3.

**Workshop – supportive work culture for ECRs.**

*Part 1—Work-life challenges in science as Early Career Researchers:*

The first part focused on the discussion about work-related stress working in research, with the key questions: *"What was the most stressful experience you experienced in your work so far? How did you overcome such a situation"*?

Participants discussed the challenges of working with ECR or experienced colleagues, depending on their perspective. Structural difficulties included separate workplaces and limited discussion time, while the hierarchical organization

**Table 1. Strategies for coping with communication challenges with co-workers and supervisors in animal experiments, according to the survey participants.**

| Category | Exemplary responses |
|---|---|
| **Standardization and documentation** | *"Clear guidelines for animal handling that everyone has to follow."*<br>*"Shared lab books to be able to read protocols and look into results of co-workers."*<br>*"Clear SOP guidelines for the whole lab."*<br>*"Fixed protocols that must be followed by each experimenter, enforcement of documentation, preferably shared files."* |
| **Team roles and responsibilities** | *"Involve PIs and professors in the discussions, not only the people that perform the experiments."*<br>*"More detailed instructions from one of my supervisors regarding what is expected, already before the task is taken up."* |
| **Information sharing and collaboration** | *"Animal experimentation should ideally be done in small teams within the lab. This way, responsibility for animals, as well as all the struggles that come with these experiments, are shared across people while avoiding having to communicate every detail to a whole lab."*<br>*"Broader meetings with other groups. "*<br>*"Taking time to have the meeting with other stakeholders (e.g., animal carer, head of the animal facility)."*<br>*"Honest reporting and accountability for the things that happen with and to the animals."* |
| **Support and training** | *"Longer training before first steps alone."*<br>*"Support and supervision during legal aspects (applications etc.)."*<br>*"Proper education of students on animal welfare and designing robust experiments."*<br>*"Training about effective communication."* |
| **Effective meeting practices** | *"Regular meetings in your group and everybody pointing out any issue they encounter at this time to find help to solve it. "*<br>*"Regular meetings with the person in charge of animal facility/ experiments and the PIs."*<br>*"Talk about the work, not only about the data."* |
| **Emotional understanding** | *"Empathy. Understanding that some days can be more stressful than usual due to other circumstances."*<br>*"Understanding and support when Stress or fears are expressed, longer training before first steps alone, support and supervision during legal aspects (applications, etc.)."*<br>*"Less critical approach to mistakes."* |

meant new ideas were not always welcomed. There was a notable difference in knowledge and experience between older scientists and early career researchers (ECRs). Approaches have been discussed to overcome difficulties, summarized as 'bridging the gap.' This combines different experiences and approaches in a complementary way. Empathy is expected toward colleagues, regardless of their training and organizational affiliation.

### Part 2- Early Career Researchers and Animal Care:

This section underlined the high demand for knowledge and skills in the area. Key questions were: "*What do you expect from Early Career Researchers/ What is the role of Early Career Researchers in understanding and recognizing the 3Rs?*"

Participants pointed out that ECRs are often open-minded individuals who carefully consider their new professional environments. By questioning existing workflows, they usually encourage their reevaluation and initiate possible process

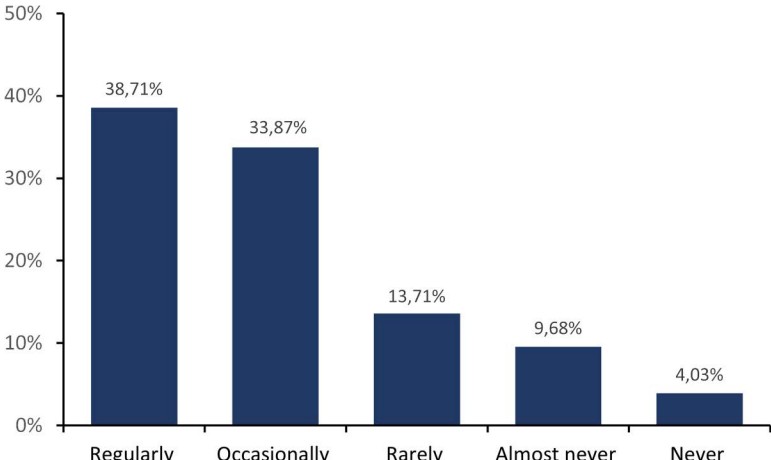

**Fig 3. Regularity of training and education of ECRs in animal care and 3R.** In this question (number 19 of the survey), participants were asked: *"How often do you receive training and education in animal care and the 3Rs (Replacement, Reduction and Refinement) from your institution?".*

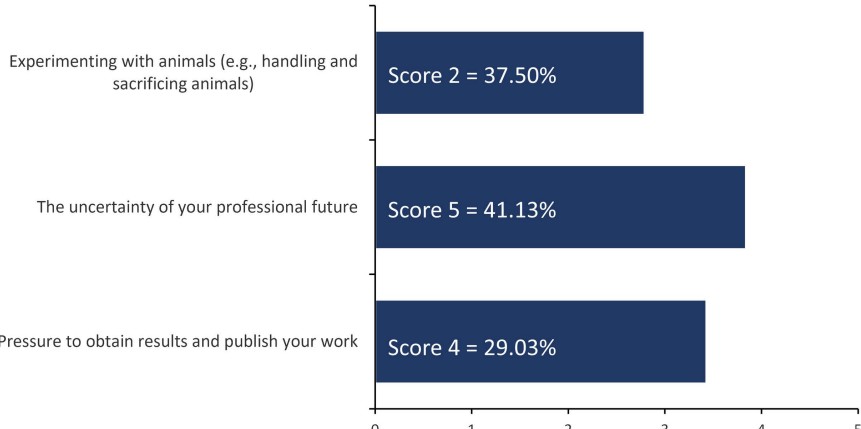

**Fig 4. Perception of three sources of stress on ECRs.** In this question (number 6 of the survey), we asked: *"How often do you experience stress regarding the following aspects?".*

updates. This is mainly because they are not yet integrated into existing hierarchical structures, and therefore, they evaluate established processes more critically. On the other hand, they state that they have a new way of thinking due to their level of training.

***Part 3- Communication with Early Career Researchers:***

This section pointed to the crucial role of communication within a positive, supportive work culture. Key questions were, *"What challenges are observed in the communication with ECRs? What practical strategies might manage such challenges?"*

In the view of our participants, the lack of structure, e.g., time for communication and trustworthy collaboration, is one of the most significant challenges. Exchange and support of each other, regardless of the existing hierarchy or experience levels, should be the aim of a supportive work culture. The hierarchical levels in an institution also lead to different levels

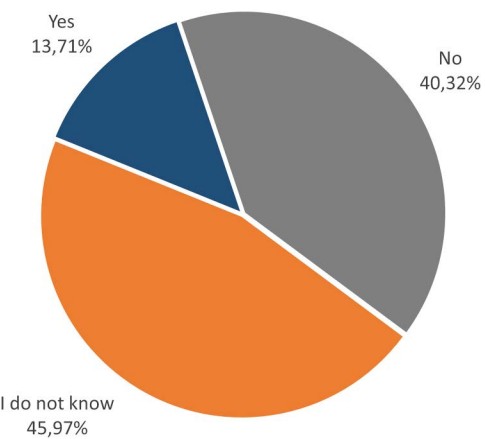

**Fig 5. Perception of the ECRs about institutional provision of tools to manage stressful situations.** In this question (8 of the survey), we asked: *"Does your institution provide tools to manage stress and improve resilience?" (Resilience here means: internal and external resources to overcome/ bounce back challenges/stress/crisis with the ability to handle this situation and grow).*

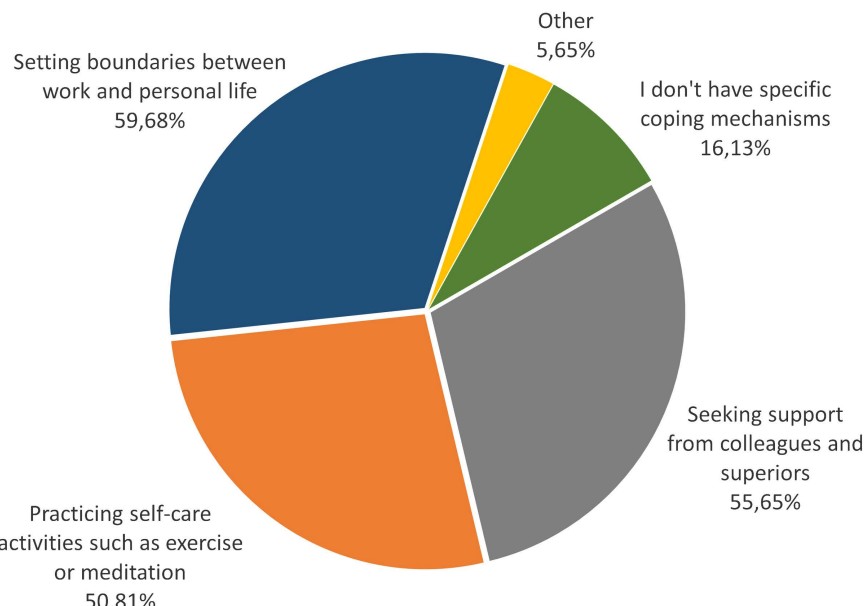

**Fig 6. Strategies to cope with stressful situations by ECRs.** In this question (9 of the survey), we asked: *"How do you currently cope with stress and challenges related to your daily work?" (Resilience here means: internal and external resources to overcome/bounce back challenges/stress/crisis with the ability to handle this situation and grow).*

of empowerment and behavior. Long-serving employees high in the hierarchy sometimes cling to their positions and their associated rights. Sticking to habits and lacking openness toward new employees also harbor potential for conflict. Sometimes, a lack of understanding of the different levels of knowledge or trust can lead to problems. Open communication was highlighted as a solution to many problems: *"A culture of listening should improve the atmosphere and create understanding and trust."* Regular meetings and follow-up strategies, shared offices, or even open office doors are frequently mentioned as simple measures to improve internal communication between ECRs and experienced colleagues.

**Table 2. Selection of written comments from participants regarding question 8 on institutional support for stress management and the improvement of resilience.**

**Question 8 – institutional support:**
*"Does your institution provide tools to manage stress and improve resilience?"*

| Answers |
| --- |
| *"Workshops (for all disciplines)"* |
| *"I know the University offers activities for personal development. But they are never tailored for people who work with animals."* |
| *"Workshops, external counseling"* |
| *"Courses, sport, counseling "* |
| *"Online seminars (optional)"* |
| *"An external service for prevention and protection at work"* |
| *"I get support in practical work from animal welfare officers to improve the workflow, thus facilitating the research, which is helpful to reduce stress."* |
| *"Stress management courses for doctoral students"* |
| *"Psychological counseling, employee training"* |
| *"There is an app"* |
| *"External Consultants"* |

**Table 3. Selection of written comments from participants in response to question 9 regarding current personal coping mechanisms for stress and challenges in daily work.**

**Question 9 on personal stress and challenges management:**
*"How do you currently cope with stress and challenges related to your daily work?"*

| Answers |
| --- |
| *"Wine"* |
| *"Family and friends"* |
| *"Discussions with friends and family about my work"* |
| *"Doing sports in the gym and outside"* |
| *"Sports, hobbies"* |

### *Part 4- Stress coping and resilience:*

The section focused on managing work-related stress through a supportive work culture and personal strategies. Key questions were: *"In which situation did you experience, or did you not experience, social support from the facility or supervisor, and what was helpful? What is your own strategy?"* The strongest feedback within this discussion point was that it has an extremely positive effect if someone affected by work-related stress feels the sentiment of *"We are all in the same boat"* and that only joint action can lead to success. In internal conflicts between two parties, an open discussion with different professionals, such as those within the animal welfare body, can also defuse tense relationships or situations and lead to a joint solution. It has also proven beneficial to use the new ideas of the ECR to reflect on and adapt established processes.

### *Part 5- Conclusion and wrap-up:*

The final section was used to exchange ideas on creating a more supportive work environment for ECRs, with a crucial focus on internal communication. The key question was: How should internal communication be changed to support ECRs? Team offices were suggested instead of individual workstations. Time for discussion is fundamental and should have fixed slots. Prioritizing work processes is key. Participants considered collegial exchange and practical work in the

institution equally important. To overcome hierarchy, there needs to be openness. Staff at all levels should recognize they have shared concerns. It was agreed that a neutral discussion between both sides is necessary because everyone learns. A trusting environment is needed so all can express their opinions freely. This group work challenged the spatial concept of a research institution. Office doors should be open at certain times to lower barriers to inquiries. The group concluded that there should be mutual listening and a culture of regular follow-up. All participants summarized the ideas and words they would like to impart to ECRs for their professional work. The main results of this group work were that ECRs should be open-minded, role-aware, and empathetic. They should have the courage to express their opinions, but also to take responsibility. ECRs should strive for teamwork that encourages open communication, regardless of the existing hierarchy.

## Discussion

### Creating a supportive work culture for ECRs: Internal communication as a key factor

Working in research, particularly with research animals, can be rewarding and challenging at the same time. Despite thriving in their careers, ECRs face high demands to deliver high-quality research outcomes and maintain the highest standards of care for research animals. Hence, there is a great need to create a supportive work culture that promotes an environment where ECRs can meet these goals while safeguarding their mental well-being. The development of a culture of care in animal research should be a focused concept for achieving this. The internationally used term 'culture of care' encompasses an approach to animal research that maximizes animal well-being as much as possible while also focusing on human well-being, as we know the two are linked. Ensuring the quality of research and a proactive, transparent communication culture are also key factors in a positive care culture [12–14]. Culture affects performance [15] and is the most critical and complex component of any organization. Internal communication plays a vital role in creating a supportive work culture. The central element of the internal communication culture is the organizational mission statement, which contains the values of an organization and aligns thinking and actions accordingly. Positive internal communication shapes employee perceptions of the organization and fosters employee engagement. Institutions and supervisors are encouraged to invest more in internal communication beyond unidirectional information transfer [16]. Multimodal information promotes effective relationships among employees, their co-workers, and organizations [17,18], which is reflected in strengthening worker engagement with strategic and research objectives [19].

Our survey results highlight the need for improving communication between ECRs and their supervisors and co-workers in animal research. The roots of the suboptimal communication include a lack of time or interest, ineffective meetings or discussions, insufficient information exchange, ambiguous instructions, conflicts and tension among team members, and interpersonal conflicts. Our workshop group discussion underlines these results, as the lack of structures to promote internal trustworthy communication plays a key role in avoiding support and the development of ECRs. Suboptimal communication results worry not only for ECRs' well-being but also for animal well-being and scientific integrity outcomes.

Optimal internal communication relies on a multimodal dynamic game in which all ECRs, supervisors, and co-workers transform the work environment into one that is respectful, stimulating, challenging, and supportive. This does not mean avoiding critical debates, scientific confrontations, frustrations, or emotional engagement [20]. As these issues are part of daily work and contribute to shaping the character of ECRs personally and professionally, it is essential to establish communication channels to introduce employees in general into the scientific system with appreciation, involving active listening, providing feedback, demonstrating care and support, creating connections and community, having open discussions on balanced expectations, and planning timelines for conducting experiments [16]. Optimal internal communication contributes to preserving ECRs' motivation and aligning with their intrinsic values [21,22]. After all, allowing members to provide insights and engage in discussions within the scientific community is a win-win for all stakeholders, including animals and science. Moreover, once ECRs feel valued and part of a team that cares for them, they can channel their extrinsic motivations into scientific, societal, and humanitarian fields [23,24], which include the care and well-being of animals

used in their experiments. A nagging question that haunts us is: How can we improve internal communication with ECRs for better well-being and science? Santos and colleagues [25] offer some recommendations that can be extrapolated to the academic and research environment. These are: 1) knowing how to listen to and respect co-workers, 2) maintaining responsiveness, 3) providing feedback and facilitating the flow of research group-related information, and 4) choosing the appropriate communication channels. We also present our toolset acquired from our learning along with this activity in the conclusions of this manuscript.

**Creating a supportive work culture for ECRs: Promoting training concepts**

The participants in the survey were asked how often training and continuing education programs on the animal care and on aspects of the 3Rs principle are provided in their scientific institutions. The background to this was regular training in the handling of laboratory animals (which is complementary to the mandatory training required to perform experiments with animals), for example, teaching new gentle methods such as tunnel handling and learning new approaches to aspects of the 3Rs. This also includes non-animal methods, such as alternative and complementary methods to animal testing. The differences in the provided training are immense. In all institutions, training is expected before commencing research activities. However, there are already initial differences here. In some cases, only theoretical knowledge is expected, while practical skills are also required in most cases. In further work, the requirements for regular additional training also differ. Surprisingly, around 40% of the institutions offer regular further and advanced training in addition to initial training (Fig 3). In some cases, no regular training is not expected at all, while other institutions require a certain number of mandatory training hours per year, ranging from 3 to 8 hours per year.

The institutions themselves often offer regular further training. In some cases, participants stated they can only access external opportunities because the institution does not provide regular training. In the regular training courses, it was often mentioned that there are no requirements for the course content. The training is usually recognized as long as aspects of animal well-being or animal care are considered. How the recognition of the training itself works was very rarely mentioned. There is no information about whether the knowledge imparted is also tested in a final exam. However, alternatives to animal testing (New Approach Methodologies), as an essential aspect of the 3Rs principle, are not regularly discussed in all scientific institutions. Further training courses on this topic do not generally take place, and if they do, then mainly in a theoretical format. The participants state that New Approach Methodologies, in particular, are discussed when they are regularly used and established in the research groups. Overall, some facilities do not appear to have a regular concept for theoretical and practical training on animal well-being and the 3Rs with mandatory participation. Due to the aforementioned uncertainty regarding whether there are any requirements, it is also possible that the survey participants are unaware of this. However, the majority of respondents stated that at least theoretical training was considered mandatory.

The legal basis for the expertise required to work with laboratory animals in the EU is Directive 2010/63/EU and specification for the work in Germany can be found in § 16 in conjunction with Annex 1 Animal Welfare Experiments Ordinance. Article 23 of this directive requires that, in addition to acquiring the necessary species-specific expertise, this expertise must be maintained and demonstrated at all times. FELASA (Federation of European Laboratory Animal Science Associations) now recommends a modular approach to competence development with defined learning outcomes. Initial training provides basic knowledge and understanding, the first step in the learning process. It involves a program of work/study leading to specific learning outcomes. Simulators, such as dummies, as the first stage of training before learning procedures on live animals, are becoming increasingly important, and many new developments have been made that are user-optimized, realistic, and affordable [26–28]. After initial training, supervised work with animals leads to a deeper understanding. This second level of training develops the in vivo skills needed to care for and work with laboratory animals responsibly and in the spirit of the '3Rs'. This can be further developed through a continuous professional development program. Competence in handling laboratory animals reduces trainee stress by enhancing confidence and skill. Proper training ensures familiarity with procedures, minimizes uncertainty, and prepares trainees for unexpected situations,

leading to fewer mistakes and greater efficiency. A structured training environment with ongoing support further promotes confidence, ultimately improving both trainee well-being and research quality.

Our survey results also emphasize our workshop outcomes: a supportive work culture aims to enable ECRs to achieve outstanding research results with the highest possible standards of animal well-being. It should provide a training concept to fulfill the demand for knowledge and skills in this area and grounds for exchange and support among one another.

### Creating a supportive work culture: Safeguarding mental well-being in ECRs

Working in animal research as a young scientist means gaining new knowledge and medical advances for society. Still, it also comes with high expectations and demands regarding their careers and caring for animals to ensure the highest possible standard of individual animal well-being. This unique situation, along with general workplace stress, can lead to impaired mental well-being in ECRs. However, safeguarding mental well-being should be a priority in occupational health management. We already know that the cost of caring contributes to the mental health of all professionals working in animal research, with the emergence of a specific phenomenon known as compassion fatigue. Burnout and secondary traumatic stress, in particular, play a role in the development of compassion fatigue. Consequently, a supportive work culture that encourages team and individual resilience is vital to combat work-related stress and safeguard professionals in animal research.

A recently published study by Young and colleagues [29] investigated the connection between professional quality of life and work metrics like retention and job satisfaction in animal research. In summary, the group found in their survey that personnel who reported higher compassion satisfaction also reported higher retention and job satisfaction. In contrast, lower job satisfaction was associated with higher burnout. Additionally, they found indications that organizational culture impacts compassion fatigue for 70% of participants (n = 118), specifically factors like feeling valued, work-life balance, training, or pay. Even though scientists were also included in this study, no special attention was paid to the situation of young scientists. Young scientists are particularly susceptible to work-related stress in research. They have to start or advance their careers; on the other hand, they take responsibility for and care for research animals. Hence, dealing with stress and building resilience are essential for ECRs. Our survey results highlight that for ECRs, the uncertainty of their professional future and the pressure to obtain results and publish work are the most important stress factors. This can certainly be explained by the fact that it is common practice, at least in Germany, for young academics not to have a permanent employment contract and for academic publications to play a decisive role in determining their future career paths. Even though providing tools to manage stress and improve resilience should be part of a supportive work culture, only a minority of our participants confirmed that this is the case in practice. Surprisingly, the most common answer was, *"I don't know if my institution provides tools."* Despite setting boundaries between work and personal life, seeking support from colleagues and supervisors is the most common strategy to cope with stress and challenges related to daily work. Our workshop outcome (part 4) also emphasizes that social support and trustworthy communication are important tools to combat work-related stress and internal conflicts. Alarmingly, we also received "drinking wine" as a coping mechanism, which is nonfunctional for combating stress but poses a high risk of alcohol addiction. Hence, our results emphasize the importance of a supportive work culture with good communication as a key factor in safeguarding mental well-being.

In advocating for a transformative shift in scientific communication culture, we emphasize the importance of prioritizing scientific rigor while fostering a nurturing and supportive environment for ECRs. By doing so, we envision cultivating a resilient and innovative scientific community that thrives on mutual support and collaboration.

### Creating a supportive work culture for ECRs: Crucial conceptual elements

A culture of care respects diverse social connections, equips individuals to cope with stress, and nurtures human-to-human relationships [30]. Such a culture not only highlights the needs of both humans and animals but also recognizes that the values and expectations of ECRs profoundly influence the quality of care provided. To create such a supportive work culture

where communication achieves a level of excellence, we posit a clear need for heightened interaction to meet the standards of high-quality scientific collaboration. Identifying and managing suboptimal internal communication is challenging, partly due to a lack of emotional openness and fear of vulnerability, differing individual communication styles, cultural backgrounds, mentoring approaches, and undefined expectations from supervisors and co-workers. Issues related to these factors, combined with the preconception that ECRs are self-motivated and do not need appreciation, significantly discourage ECRs from communicating effectively and are intrinsically linked to demotivation and lower performance. This is critical in research environments, as demotivation stems not only from job abandonment but also from the careless treatment of colleagues, animals, and science, which requires intense work and investment in improving internal communication.

Listed below are our key targets to create a supportive work culture where ECRs thrive and safeguard their mental well-being.

**Enhancing Communication Strategies among ECRs and Senior Researchers and Staff:**

1. *Establish Communication Channels:*

   - Incentivize ECRs to create and conduct dedicated communication channels, such as newsletters, intranet platforms, or regular meetings, to disseminate information on well-being within research and acquire skills on the 3Rs.

   - Promote accessibility and user-friendly interfaces for emotional support in work environments, involving all the team staff.

2. *Training in Effective Communication:*

   - Provide workshops or training sessions on effective communication strategies for ECRs, focusing on open dialogue.

   - Encourage the participation of ECRs in institutional programs focused on communication and training in leadership.

3. *Regular Feedback Mechanisms:*

   - Implement an anonymous feedback system where early-career researchers can provide input on animal care practices and research methodologies.

   - Foster an environment where constructive feedback is valued and practically implemented for continuous improvement. This does not mean all the initiatives must be implemented, but they should at least be discussed.

4. *Cross-Functional Teams:*

   - Encourage interdisciplinary collaboration by forming cross-functional teams that bring together researchers, animal care staff, and 3Rs experts through intra-institutional fairs and events.

   - Promote events that foster a culture of knowledge-sharing aimed at mutual understanding between different departments.

**Understanding Cultural Differences in the Laboratory and Research Environment:**
*Diversity and Inclusion Training:*

- Conduct training programs to raise awareness of cultural differences and promote inclusivity in the workplace.

- Provide resources for researchers to educate themselves on cultural nuances that affect animal care practices and research methodologies.

*Cultural Sensitivity Workshops:*

- Organize workshops to enhance cultural sensitivity, emphasizing the importance of understanding diverse perspectives in human communication approaches, failure acceptance, and trust-giving patterns.

- Establish guidelines that address cultural considerations in implementing animal care and 3Rs initiatives.

*Diversity in Research Teams:*

- Encourage diverse recruitment practices to create research teams that reflect a range of cultural backgrounds.

- Foster an inclusive atmosphere where individuals feel comfortable expressing their perspectives.

**Implementation of Mentorship and Peer Coaching Programs:**

*Structured Mentorship Programs:*

- Establish formal mentorship programs pairing senior researchers with early-career researchers.

- Provide guidelines for mentors to offer both instrumental support (guidance on research techniques) and emotional support (coping with challenges in animal research).

*Peer Coaching Initiatives:*

- Facilitate peer coaching programs where ECRs can share their concerns, experiences, best practices, and challenges related to animal care, research, and implementing the 3Rs.

- Encourage regular check-ins among peers to create a supportive network within the research community.

*Structured Peer Feedback Sessions:*

- Facilitate regular and exclusive peer feedback sessions within research teams. This will allow researchers to provide constructive feedback to their peers regarding their ideas, questions, and concerns about animal research and implementing new methodologies.

- Establish a supportive framework for receiving and incorporating feedback.

**Strengthening Institutional Support Systems:**

*3Rs*

• Encourage employees to be open to new 3R approaches in their daily work and provide incentives, such as rewarding internal 3R projects.

*Well-being Resources:*

- Offer ECRs mental health and well-being resources, including counseling services and stress management programs.

- Promote a healthy work-life balance by establishing policies that support flexible working arrangements when feasible.

*Career Development Opportunities:*

- Provide funding and resources for professional development, such as attending conferences, workshops, and courses related to animal care, research, and implementing the 3Rs.

- Create pathways for career advancement within the institution, recognizing and rewarding research excellence.

*Advocacy for Researchers:*

- Advocate for researchers at the institutional and legal levels (e.g., regarding the legal consequences of unintentional misconduct with animals), ensuring that management hears their concerns and needs.

- Establish a liaison role that acts as a bridge between researchers and administrative entities to address issues promptly.

*Flexibility and Support for Care Responsibilities:*

- Acknowledge and support researchers with caregiving responsibilities by offering flexible work schedules, parental leave policies, and on-site childcare facilities where feasible.

- Ensure that these policies are communicated clearly and implemented consistently.

## Limitations of our study

Despite the insights that our study provides, it is essential to acknowledge certain limitations in assessing communication that may impact the generalizability of our findings. The survey-based nature of our research introduces potential response biases, as participants' perceptions and experiences are inherently subjective. Furthermore, the cross-sectional design restricts our ability to establish causal relationships or capture the dynamic nature of communication over time. It is noteworthy that we did not extensively explore organizational and institutional factors influencing communication, which could be the subject of future research. Given the escalating emphasis on interdisciplinary collaborations, future research should explore how communication dynamics vary across different scientific disciplines and impact the career trajectories of ECRs working in animal research. Comparative studies between institutions and countries could offer valuable benchmarks for best practices in fostering effective internal communication. The study was conducted in Germany and comments may be biased to the local situation. We recommend to extend the qualitative approaches to other countries. Additionally, integrating technology and innovative communication tools may open new avenues for improving information exchange and collaboration among ECRs.

## Conclusion

This paper explored the multifaceted challenges ECRs face in animal experimentation in academic and industrial settings. The identified challenges, including suboptimal working conditions, a lack of consensus on 3Rs application and training, and inadequate stress coping strategies, contribute to unintended harm to animals and jeopardize the overall scientific rigor of experiments. This study underscores the urgency of addressing these issues directly, fostering a paradigm shift towards a more supportive and inclusive environment for ECRs.

## Supporting information

**S1 Table.  Survey questions.**
(XLS)

**S2 Table.  Survey responses.**
(XLS)

## Acknowledgments

We extend our heartfelt thanks to survey participants for enhancing the research with valuable insights and to workshop contributors for enriching academic discussions with their time and expertise.

## Author contributions

**Conceptualization:** Fernando Gonzalez-Uarquin, Fabienne Ferrara, Nadine Baumgart, Sabine Juliane Bischoff.

**Data curation:** Fernando Gonzalez-Uarquin, Fabienne Ferrara, Sabine Juliane Bischoff.

**Formal analysis:** Fernando Gonzalez-Uarquin, Fabienne Ferrara, Sabine Juliane Bischoff.

**Funding acquisition:** Nadine Baumgart.

**Investigation:** Fernando Gonzalez-Uarquin, Fabienne Ferrara, Sabine Juliane Bischoff.

**Methodology:** Fernando Gonzalez-Uarquin, Fabienne Ferrara, Sabine Juliane Bischoff.

**Project administration:** Fernando Gonzalez-Uarquin.

**Resources:** Fabienne Ferrara, Jan Baumgart, Sabine Juliane Bischoff.

**Supervision:** Fernando Gonzalez-Uarquin, Fabienne Ferrara, Jan Baumgart, Sabine Juliane Bischoff.

**Validation:** Fernando Gonzalez-Uarquin, Fabienne Ferrara, Nadine Baumgart, Jan Baumgart, Sabine Juliane Bischoff.

**Visualization:** Fernando Gonzalez-Uarquin, Fabienne Ferrara, Nadine Baumgart, Jan Baumgart, Sabine Juliane Bischoff.

**Writing – original draft:** Fernando Gonzalez-Uarquin, Fabienne Ferrara, Sabine Juliane Bischoff.

**Writing – review & editing:** Fernando Gonzalez-Uarquin, Fabienne Ferrara, Nadine Baumgart, Jan Baumgart, Sabine Juliane Bischoff.

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
