## [Decision Letter · Decision Letter 0]

22 Jul 2025

Dear Dr. Gonzalez-Uarquin,

Thank you for submitting your manuscript to PLOS ONE. After careful consideration, we feel that it has merit but does not fully meet PLOS ONE’s publication criteria as it currently stands. Therefore, we invite you to submit a revised version of the manuscript that addresses the points raised during the review process.

We look forward to receiving your revised manuscript.

Kind regards,

Sourav Roy

Academic Editor

PLOS ONE

Journal Requirements:

2. Please include captions for your Supporting Information files at the end of your manuscript, and update any in-text citations to match accordingly. Please see our Supporting Information guidelines for more information: http://journals.plos.org/plosone/s/supporting-information .

Additional Editor Comments:

Please revise the manuscript per the reviewers' suggestions. It is important to address every point raised by each of the three reviewers, for this manuscript to be considered for publication.

Reviewers' comments:

Reviewer's Responses to Questions

**Comments to the Author**

1. Is the manuscript technically sound, and do the data support the conclusions?

Reviewer #1: Yes

Reviewer #2: Yes

Reviewer #3: Yes

2. Has the statistical analysis been performed appropriately and rigorously?

Reviewer #1: Yes

Reviewer #2: N/A

Reviewer #3: N/A

3. Have the authors made all data underlying the findings in their manuscript fully available?

Reviewer #1: Yes

Reviewer #2: Yes

Reviewer #3: Yes

4. Is the manuscript presented in an intelligible fashion and written in standard English?

Reviewer #1: Yes

Reviewer #2: Yes

Reviewer #3: Yes

Reviewer #1: The work presented is a relevant basis for improving the environment and conditions in which animal research is carried out, first and foremost in terms of the welfare of researchers. The framework of the survey is restricted to a limited geographical area, and this aspect should be considered in the title.

Reviewer #2: The research is a good one. However, there are few corrections as outlined below:

*In the Abstract section, mention is made of 3Rs without stating what the 3Rs represent. Even though the full meaning is given in the main body of the work, it is important that this be stated at first mention, especially in the Abstarct section as this is the part a reader would read first.

*Check line 239 under Results section and correct accordinglhy. What is written ther is (refinement, replacement and refinement). Refinement is repeated twice.

* Line 391 under the Discussion section should be "employees" and not "employes"

Reviewer #3: The manuscript deals with the possible health and mental risks for early career researchers (ECRs) who are involved in animal experiments in their research. A brief introduction highlights the role of ECRs, identifies challenges for this group of scientists and speculates on possible causes for these challenges. The experimental part of the study includes a survey with 124 ECRs and conducting a workshop on the matter. The collected responses were analyzed descriptively and provide an overview of the degree of stress, potential causes and applied coping mechanisms. The results are discussed in detail and a number of suggestions for improving the situation are provided.

The present study deals with an important aspect of research that can have a major impact on the individual careers of ECRs, scientific work with animals and the results of such studies. The scientific component of this study may be lower than in other publications, but this is outweighed by its social and scientific relevance. I therefore consider it essential to publish the results of the study.

The article is well written and comprehensible and is particularly convincing due to the thoroughness of the discussion. I only have a few (almost picky) suggestions for changes.

Line 137

12 participants in 3 groups of 3 doesn’t add up – please correct numbers

Line 297

This section begins somewhat unfavorably and needs to be formulated clearly: in the methods section participants were described as homogenous group (senior staff, older than 40) different from ECRs - however, the question posed in the workshop part 1 suggests the perspective of the ECRs. Perhaps the text should be reworded here so that it is clear who comes to this conclusion.

Line 339

ECS – wrong abbreviation

Line 548

change “labor” to “working” or “laboratory”

**Do you want your identity to be public for this peer review?** For information about this choice, including consent withdrawal, please see our Privacy Policy

Reviewer #1: No

Reviewer #2: No

Reviewer #3: **Yes: ** Paul Mieske

---

## [Author Response · Author response to Decision Letter 1]

5 Aug 2025

PONE-D-25-09004

Fostering culture of care for early career researchers – building a trustful environment: Insights from a German perspective

PLOS ONE

Dear Dr. Sourav Roy,

We would like to thank you and the reviewers for the time and effort dedicated to evaluating our manuscript. We greatly appreciate the thoughtful and constructive feedback provided, which has helped us improve the clarity and quality of our work.

Please find below our detailed responses to the comments raised by the editorial board, Reviewer 1, Reviewer 2, and Reviewer 3. We have addressed each point carefully and made the corresponding revisions in the manuscript, which are highlighted for ease of reference. Where appropriate, we have also included a brief justification for our approach or clarified our intentions.

We hope the revised version meets the expectations of the journal and look forward to your further consideration.

Sincerely,

Dr. Fernando Gonzalez Uarquin

Editorial Comments

Additional Editor Comments:

Please revise the manuscript per the reviewers' suggestions. It is important to address every point raised by each of the three reviewers, for this manuscript to be considered for publication.

Thank you for your comments. We have addressed all the points mentioned by the reviewers. In addition, we amended the affiliation of the authors as we have realized of some mistakes (lines 10 – 18 of the revised manuscript with track changes).

In addition, we fixed the space between headings and paragraphs in accordance with the guidelines.

Reviewers’ comments

Reviewer #1:

The work presented is a relevant basis for improving the environment and conditions in which animal research is carried out, first and foremost in terms of the welfare of researchers. The framework of the survey is restricted to a limited geographical area, and this aspect should be considered in the title.

Thank you for your relevant feedback. The comment about the title is a valid point, and we have revised the title accordingly to reflect the limited geographical scope of the survey: “Fostering culture of care for early career researchers – building a trustful environment: Insights from a German perspective.”

Reviewer #2:

The research is a good one. However, there are few corrections as outlined below:

1) In the Abstract section, mention is made of 3Rs without stating what the 3Rs represent. Even though the full meaning is given in the main body of the work, it is important that this be stated at first mention, especially in the Abstarct section as this is the part a reader would read first.

We thank for your comment and amended the fragment of the abstract accordingly: “…3Rs (replacement, reduction, and refinement)…”

2) Check line 239 under Results section and correct accordinglhy. What is written ther is (refinement, replacement and refinement). Refinement is repeated twice.

Thank you for pointing out this oversight (line 239 of the revised manuscript with track changes). We have corrected it according to your comment.

3) Line 391 under the Discussion section should be "employees" and not "employes"

We apologize for the typo. The word employes was changed by employees in line 392 of the revised manuscript with track changes of the tracked document.

Reviewer #3:

The manuscript deals with the possible health and mental risks for early career researchers (ECRs) who are involved in animal experiments in their research. A brief introduction highlights the role of ECRs, identifies challenges for this group of scientists and speculates on possible causes for these challenges. The experimental part of the study includes a survey with 124 ECRs and conducting a workshop on the matter. The collected responses were analyzed descriptively and provide an overview of the degree of stress, potential causes and applied coping mechanisms. The results are discussed in detail and a number of suggestions for improving the situation are provided.

The present study deals with an important aspect of research that can have a major impact on the individual careers of ECRs, scientific work with animals and the results of such studies. The scientific component of this study may be lower than in other publications, but this is outweighed by its social and scientific relevance. I therefore consider it essential to publish the results of the study.

The article is well written and comprehensible and is particularly convincing due to the thoroughness of the discussion. I only have a few (almost picky) suggestions for changes.

1) Line 137: 12 participants in 3 groups of 3 doesn’t add up – please correct numbers

We apologize for the mistake. The word “three” was changed by “four” employees in line 137 of the revised manuscript with track changes.

2) Line 297: This section begins somewhat unfavorably and needs to be formulated clearly: in the methods section participants were described as homogenous group (senior staff, older than 40) different from ECRs - however, the question posed in the workshop part 1 suggests the perspective of the ECRs. Perhaps the text should be reworded here so that it is clear who comes to this conclusion.

Thank you very much for your comment. We reworded accordingly (from line 297 of the revised manuscript with track changes): “Participants discussed the challenges of working with ECR or experienced colleagues, depending on their perspective. Structural difficulties included separate workplaces and limited discussion time, while the hierarchical organization meant new ideas were not always welcomed.”

3) Line 339: ECS – wrong abbreviation

Thank you very much for catching this typo, “ECS” was changed by “ECR” employees in line 339 of the revised manuscript with track changes.

4) Line 548: change “labor” to “working” or “laboratory”

Thank you for pointing out this oversight in what is now line 556 of the revised manuscript with track changes. We have corrected it according to your comment.

---

## [Editor Report · Decision Letter 1]

20 Aug 2025

Fostering culture of care for early career researchers – building a trustful environment: Insights from a German perspective

PONE-D-25-09004R1

Dear Dr. Gonzalez-Uraquin,

We’re pleased to inform you that your manuscript has been judged scientifically suitable for publication and will be formally accepted for publication once it meets all outstanding technical requirements.

Within one week, you’ll receive an email detailing the required amendments. When these have been addressed, you’ll receive a formal acceptance letter, and your manuscript will be scheduled for publication.

Kind regards,

Sourav Roy

Academic Editor

PLOS ONE

Additional Editor Comments (optional):

Please change the title to:  Fostering a culture of care for early career researchers – building a trustworthy environment: Insights from a German
---

## [Editor Report · Acceptance letter]

PONE-D-25-09004R1

PLOS ONE

Dear Dr. Gonzalez-Uarquin,

I'm pleased to inform you that your manuscript has been deemed suitable for publication in PLOS ONE. Congratulations! Your manuscript is now being handed over to our production team.

Kind regards,

on behalf of

Dr. Sourav Roy

Academic Editor

PLOS ONE